# Antiviral and Virucidal Activities of *Uncaria tomentosa* (Cat’s Claw) against the Chikungunya Virus

**DOI:** 10.3390/v16030369

**Published:** 2024-02-27

**Authors:** Raquel Curtinhas de Lima, Ligia Maria Marino Valente, Débora Familiar Macedo, Luzia Maria de-Oliveira-Pinto, Flavia Barreto dos Santos, José Luiz Mazzei, Antonio Carlos Siani, Priscila Conrado Guerra Nunes, Elzinandes Leal de Azeredo

**Affiliations:** 1Laboratório das Interações Vírus Hospedeiros, Instituto Oswaldo Cruz, Rio de Janeiro 21040-900, Brazil; raquellima@aluno.fiocruz.br (R.C.d.L.); deborafamiliar@gmail.com (D.F.M.); lpinto@ioc.fiocruz.br (L.M.d.-O.-P.); flaviab@ioc.fiocruz.br (F.B.d.S.); priscila.nunes87@gmail.com (P.C.G.N.); 2Instituto de Química, Universidade Federal do Rio de Janeiro, Av. Athos da Silveira Ramos, 149, Rio de Janeiro 21941-909, Brazil; valente@iq.ufrj.br; 3Laboratório de Tecnologia para Biodiversidade em Saúde, Instituto de Tecnologia de Fármacos, Fundação Oswaldo Cruz, Rio de Janeiro 21041-250, Brazil; mazzei@ymail.com (J.L.M.); antonio.siani@fiocruz.br (A.C.S.)

**Keywords:** Chikungunya, *Uncaria tomentosa*, antiviral and virucidal effects, cat’s claw

## Abstract

*Uncaria tomentosa* (UT) is a medicinal plant popularly known as cat’s claw belonging to the Rubiaceae family that has been reported to display antiviral and anti-inflammatory activities. The chikungunya virus (CHIKV) outbreaks constitute a Brazilian public health concern. CHIKV infection develops an abrupt onset of fever, usually accompanied by a skin rash, besides incapacitating polyarthralgia. There is no vaccine available or treatment for CHIKV infection. The present study evaluates the hydroalcoholic extract of UT bark as a potential antiviral against CHIKV. The in vitro antiviral activity of the UT extract against the Brazilian CHIKV strain was assessed using quantitative reverse transcription polymerase chain reaction, flow cytometry, and plaque assay. Results obtained demonstrated that UT inhibits CHIKV infection in a dose-dependent manner. At the non-cytotoxic concentration of 100 µg/mL, UT exhibited antiviral activity above 90% as determined by plaque reduction assay, and it reduced the viral cytopathic effect. Similarly, a significant virucidal effect of 100 µg/mL UT was observed after 24 and 48 h post-infection. This is the first report on the antiviral activity of UT against CHIKV infection, and the data presented here suggests UT as a potential antiviral to treat CHIKV infection.

## 1. Introduction

Chikungunya virus (CHIKV) is an arthropod-borne arthritogenic virus belonging to the *Togaviridae* family, *Alphavirus* genus [1], and the etiologic agent of chikungunya fever [2]. CHIKV is an enveloped virus with icosahedral symmetry and approximately 70 nm in diameter. Its genome is composed of single-stranded RNA of positive polarity, which contains approximately 11.8 kb [3].

Initially, chikungunya cases were limited to outbreaks in Africa and Asia; however, in 2005, an intense epidemic reached the Indian Ocean islands, affecting more than 300,000 people [4,5]. In 2013, the virus was introduced in the Americas and has caused more than 3.5 million cases ever since. More specifically, in Brazil, annual epidemics have been reported since then [6]. The co-circulation of different arboviruses hinders clinical-epidemiological diagnosis due to the non-specificity of most symptoms, which makes laboratory diagnosis important for correct patient management and epidemiological surveillance [7,8].

About 80 to 90% of CHIKV infections are symptomatic [9,10], although recent studies indicate that the percentage of asymptomatic patients may be higher [11]. Chikungunya fever is an acute illness with a sudden onset. In typical symptomatic cases, the common symptoms are high fever (>39 °C), headaches, myalgia, skin rash, and intense polyarthralgia, the most characteristic feature of chikungunya cases [12]. The disease tends to be self-limiting, evolving to cure in approximately two weeks; however, in 48 to 80% of arthritogenic patients, the symptoms persist for months or years, evolving into a chronic phase. Outbreaks from different regions showed a similar clinical outcome, compromising the joints in the long term [5,13,14,15,16]. Atypical manifestations such as skin pigmentation, ulcers, chest pain, and ocular and neurological symptoms are described during the acute phase [17,18,19], and lethal complications such as myocarditis, hepatitis, and encephalitis have also been reported [20,21]. Children, who tend to be more vulnerable to central nervous system infections [8], present more neurologic manifestations when compared to adults [22]. Hemorrhagic and dermatological manifestations are also more common in this age group [22,23].

In its urban cycle, CHIKV is transmitted by female mosquitoes *Aedes aegypti* and *Ae. albopictus* [24]. The virus is inoculated into the skin during blood meal and infects the local cells, such as epithelial cells, fibroblasts, and macrophages. This initial replication generates an intense immune response that induces the expression of antiviral and inflammatory genes [25]. Still, the initial response is not sufficient to contain the virus, which interacts with other cells, such as Langerhans cells, that migrate to the lymph nodes and then spread to organs such as the liver and spleen, considered primary sites of replication, and then to muscle, brain, and articulation [26]. It may also be inoculated directly in the blood circulation [26]. It is commonly accepted that persistent arthralgia in chikungunya disease is the result of the host’s inflammatory response, among other factors. We and others have shown increased concentrations of cytokines and chemokines such as Tumor Necrosis Factor-α (TNF-α), Interferon-γ (IFN-γ), Interleukin-6 (IL-6), IL-10, Chemokine (C-X-C motif) ligand 9 (CXCL9/MIG), Monocyte chemoattractant protein 1 (CCL2/MCP-1), Interferon gamma-induced protein 10 (CXCL10/IP10) and C-X-C Motif Chemokine Ligand 8 (CXCL8/IL-8) in chikungunya infected patients. In addition, higher levels of CCL2/MCP-1, IL-6, and Macrophage inflammatory protein-1 (CCL4/MIP-1β) and decreased levels of Regulated upon Activation, Normal T-cell Expressed and Secreted (CCL5/RANTES) are observed in the chronic phase of infection and arthritic cases [27,28,29,30,31].

*Uncaria tomentosa* (Willd.) D.C (UT), also known as cat’s claw, uña-de-gato or unha-de-gato, is a medicinal plant endemic to the tropical forests of Central and South America, used by Peruvian and Brazilian tribes such as Asháninka, Boras, and Kaíapor [32], being included in the third volume of World Health Organization (WHO) monographs on selected medicinal plants, a guideline with specifications for the most widely used medicinal plants [33]. Although used in the treatment of many conditions such as abscess, arthritis, asthma, and cancer, its anti-inflammatory activities are the most widely reported [32]. Indeed, we reported the antiviral and immunomodulatory properties of UT and *Uncaria guianensis* (UG) in dengue-infected cells in vitro [34,35,36]. The immunomodulatory effects were observed by the low production of TNF-α [34], macrophage migration inhibitory factor (MIF), IL-6, and CXCL8/IL-8 production [36]. Furthermore, treatment with UT alkaloid fraction reduced significantly the vascular permeability in endothelial cells infected with dengue [35].

There is no specific antiviral medication. Currently, chikungunya fever is treated with medications to alleviate symptoms, for example, pain and fever. The use of corticosteroids and immunosuppressants is indicated in patients with moderate to severe pain after the acute and chronic phases, but nonetheless, a range of severe side effects are reported [12]. There is a need for alternative therapies, especially for chronic patients, and natural products are a potential source of antivirals against vector-borne viruses in risk areas. Finally, here we investigated the antiviral effects of the hydroalcoholic extract of UT bark against CHIKV infection in vitro. Our results revealed for the first time that UT exhibited antiviral and virucidal activity against CHIKV infection, suggesting UT as a potential anti-CHIKV agent.

## 2. Materials and Methods

### 2.1. Plant Extract

The hydroalcoholic ethanol/water 1:1 extract was previously obtained [37] from the stem barks of UT wild specimens collected in Cruzeiro do Sul, Acre, Brazil (donated by Biosapiens Co., Brazil—voucher data in Miranda and collaborators) [38]. Legal access to the Brazilian genetic heritage component is registered in the SisGen platform under number AE8FE55.

### 2.2. Preparation of the Hydroalcoholic Extract Solution of Uncaria tomentosa

A stock solution of 75 mg of extract per mL (mg/mL) was prepared using dimethylsulfoxide (DMSO, Nova Biotecnologia, Cotia, SP, Brazil), followed by the preparation of an intermediate solution of 1 mg/mL in culture-supplemented medium 199 (Life Technologies, Carlsbad, CA, USA). Working solutions were prepared with serial dilutions of the intermediate one in a medium supplemented with 2% fetal bovine serum (FBS, Life Technologies) in a 1:1 ratio starting at 200 µg/mL.

### 2.3. Cell Line

Vero cells (African green monkey kidney cells, CCL-81) were cultured in 199 medium containing L-glutamine (Thermo Fisher Scientific, Waltham, MA, USA) and supplemented with 2% FBS, 1% penicillin/streptomycin (Life Technologies), and 0.1% fungizone (Life Technologies). Cells were maintained in the incubator at 37 °C and 5% CO_2_ in 75 cm^2^ flasks until seeding time.

### 2.4. Virus Stock and Infection

The viral stock was produced using a CHIKV isolate from a serum sample collected from a patient during the 2016 chikungunya fever epidemic in Rio de Janeiro, Brazil. CHIKV diagnosis was confirmed using RT-qPCR, and its genome was sequenced and identified as East-Central-South African (ECSA) genotype (EJS20/BR/RJ/2016, Genbank accession number PP049070). The diluted viral isolate was inoculated in Vero cell culture and incubated for 1 h at 37 °C and 5% CO_2_ for viral adsorption. Subsequently, 2% FBS-supplemented 199 cell medium was added, and cells were incubated for 2 days in the same conditions. The supernatant was collected and stored at −80 °C, and the titer of viral stock was determined by plaque assay.

### 2.5. Cell Viability Assay

The viability of Vero cells in the presence of the hydroalcoholic extract of UT was assessed using MTT (3-[4,5-dimethylthiazol-2-yl]-2,5-diphenyl tetrazolium bromide) assay (Invitrogen, Waltham, MA, USA) as per the manufacturer’s instructions. Briefly, cells were seeded at the density of 10^5^ cells/mL in 96 well plates and incubated at 37 °C and 5% CO_2_ for 24 h. After the formation of the confluent monolayer, the 199 medium was removed, and distinct concentrations of extracts were added in 6 replicates and incubated for 24, 48, and 72 h. As positive and negative controls, cells were treated with 3% Tween and culture medium, respectively. In addition, to assess the DMSO cytotoxicity used in the extract preparation, the highest DMSO concentration found in the dilutions (0.4%) was tested (vehicle control). Optical densities (OD) were determined by the EZ Read 400 Microplate Reader (Biochrom, Holliston, MA, USA) at 570 nm. The viability was calculated considering the control of untreated cells as 100% viability.

### 2.6. Antiviral Assay

To investigate the antiviral activity of the hydroalcoholic extract of UT against CHIKV, Vero cells were seeded at the density of 1 × 10^5^ cell/mL in 96 well plates and incubated at 37 °C and 5% CO_2_ for 24 h until the formation of the confluent monolayer. Cells were infected with a multiplicity of infection (MOI) of 0.01, and following infection, the inoculum was removed from wells and replaced by 200 µL of the maximum non-cytotoxic concentrations of the extract and its dilutions in duplicates. In addition, CHIKV-infected cells were treated with vehicle control at 0.4% (DMSO). Supernatants and cell lysates were collected 24, 48, and 72 h post-infection for viral quantification. CHIKV infection was evaluated using plaque assay, reverse transcription-quantitative polymerase chain reaction method (RT-qPCR), and flow cytometry.

### 2.7. Virucidal Assay

To investigate the direct action of the extract on the viral particle, we performed the virucidal assay, according to Santos et al. (2021) [39]. Vero cells were seeded at the density of 1 × 10^5^ cell/mL in 96 well plates and incubated at 37 °C and 5% CO_2_ for 24 h. The CHIKV inoculum at an MOI of 0.01 was treated with non-toxic concentrations of extracts and vehicle control at 0.4% (DMSO) in polypropylene tubes for 1 h at 37 °C. Medium-treated viruses were used as a viral replication control. Subsequently, the treated inoculums were added to cell monolayers and incubated at 37 °C and 5% CO_2_. After 1 h of adsorption, the inoculum was removed and replaced by 200 µL of 2% FBS-supplemented 199 medium. The culture supernatants were collected after 24, 48, and 72 h and stored at −80 °C for RNA quantification and virus titration using real-time RT-qPCR and plaque assay.

### 2.8. RNA Extraction

Intracellular RNA was extracted using an RNeasy mini kit (Qiagen, Hilden, North Rhine-Westphalia, Germany). After the time points mentioned, supernatant was collected, cells were washed with PBS (phosphate-buffered saline), and 50 µL of trypsin with 2% EDTA (ethylenediaminetetraacetic acid) was added to each well to detach them. After 3 min incubation at 37 °C and 5% CO_2_, trypsin was inactivated using 150 µL of 199 culture medium supplemented with 10% FBS, cells were washed and lysed, and RNA extraction was performed according to the manufacturer’s instructions. Culture supernatant RNA was extracted using a QIAamp^®^ Viral RNA kit (Qiagen).

### 2.9. RT-qPCR Quantification of Viral RNA

After extraction, RNA from culture supernatants and cell lysate were submitted to RT-qPCR for CHIKV RNA quantification. Primers and probes described by Lanciotti et al. (2007), with an analytical sensitivity of 0.3 PFU [40], and GoTaq^®^ Probe 1-Step RT-qPCR System (Promega, Madison, WI, USA) were used. Reverse transcription was set at 45 °C for 15 min, and the polymerase enzyme was activated in one cycle at 95 °C for 2 min, followed by 45 amplification cycles at 95 °C for 15 s and 60 °C for 1 min. Viral RNA was quantified using a standard curve with an already-known viral titer. The reaction was performed in QIAquant 96 with a 5plex detection system thermal cycler (Qiagen) and analyzed with QIAquant 96 Software 1.0.3.0. (Qiagen).

### 2.10. Flow Cytometry Analysis for CHIKV Antigen Detection

Vero cells were CHIKV infected (MOI 0.01) and treated with 50 and 100 µg/mL of UT at 24 h or vehicle control at 0.4% (DMSO). Cells were detached with trypsin–EDTA solution, centrifuged at 450× *g* for 5 min, and resuspended in 200 μL of cold buffer (PBS, 2% FBS, and 1 mM EDTA). For viability assessment, the cells were stained with viability Dye FITC (Thermo Fisher Scientific, Waltham, MA, USA) according to the manufacturer’s recommendations. Before antigen labeling, cells were incubated in Cytofix/Cytoperm solution (Cytofix/Cytoperm Kit, BD Biosciences, Franklin Lakes, NJ, USA) at 4 °C for 20 min and then were washed with Perm/Wash Buffer (Cytofix/Cytoperm Kit). Intracellular staining was performed during the permeabilization step using Alexa 647-conjugated anti-CHIKV antibody or isotype (1:50 dilution, MyBiosource, San Diego, CA, USA) for 45 min at 4 °C. Cells were analyzed using a CytoFLEX cytometer (Beckman Coulter, Brea, CA, USA). FlowJo software version 10 (FlowJo, LLC, Ashland, OR, USA) was used in the flow cytometry data.

### 2.11. Viral Quantification by Plaque Assay

Culture supernatants collected during antiviral and virucidal assays were tittered using plaque assay. Briefly, 2 × 10^5^ Vero cells were seeded in 12-well plates and incubated at 37 °C and 5% CO_2_ for 24 h. Ten-fold serial dilutions of the supernatants collected in the previously described assays were performed in 2% FBS-supplemented 199 medium. Subsequently, 100 µL of inoculum was added to the Vero monolayers in duplicates and incubated at 37 °C and 5% CO_2_ for 1 h for adsorption. After the incubation, a semi-solid medium (0.3% agarose in 199 medium) was added to each well, and cells were incubated at 37 °C and 5% CO_2_. After 30 h, cells were fixed with formaldehyde 8% overnight, washed in running water, and then stained with 1 mL of crystal violet for 2 h. The dye was washed off and the plaques were counted after the plates dried.

### 2.12. Statistical Analysis

Data are presented as the mean ±standard deviation (SD) and were analyzed using one-way ANOVA with Turkey’s multiple comparisons linear regression analysis, directly or after conversion to the logarithm of the doses. The half maximal effective concentration (EC50) was calculated using non-linear regression analysis. All data and statistical analyses were carried out using the GraphPad Prism 6.0 software (La Jolla, CA, USA). Results were considered statistically significant if *p* < 0.05.

## 3. Results

### 3.1. Alkaloid Content and Profile of the Hydroalcoholic Extract of Uncaria tomentosa

The alkaloid content and profile of the hydroalcoholic extract batch used in this work have been previously reported [41]. Applying previously described methods [34,41], UT-derived samples were obtained from stem barks. The total alkaloid content in the crude extract was calculated as 29.1 mg/g (±1%), and the six pentacyclic oxindole alkaloids considered as the UT marker have been found (Figure 1).

### 3.2. Vero Cell Viability in the Presence of Uncaria tomentosa Hydroalcoholic Extract

The cytotoxicity of different concentrations of UT extract was assessed by treating Vero cells for up to 72 h through the MTT assay. Untreated cells were used as a reference for 100% viability, and concentrations that maintained viability lower than 80% in all three time points were considered cytotoxic. After 24 h, cells treated with 200 µg/mL had 75.3% viability, indicating cytotoxic properties of the extract in this concentration. In 48 h treatment, cell viability was higher than 90% in all tested concentrations, while 72 h treatment with 200 µg/mL decreased viability to 75% once again. Tween at 3% was included as cytotoxicity control. The tested concentrations of 6 to 100 µg, as well as 0.4% of DMSO (vehicle control), kept cell viability above 80% in all time points (Figure 2) and were, therefore, used in the antiviral activity investigation assays. Cell viability was reduced when Vero cells were treated with higher amounts (200 µg/mL) of extract at 24 h and 72 h as well.

### 3.3. The Brazilian CHIKV Isolate

Several in vitro studies have successfully used the Vero cell lineage since they are very susceptible to CHIKV infection even at low MOI at 48 h post-infection (p.i) [42,43,44]. Initially, we determined the ECSA strain cytotoxicity profile after 48 h p.i. Vero cells were infected at increasing MOI (1, 0.1, and 0.01) and then tested for cell viability. As demonstrated in Appendix A, infected Vero cells at MOI 1 resulted in lower viable cells, and the in vitro experiments were conducted at low MOI (0.01). The growth kinetics of the ECSA CHIKV strain was analyzed by quantitation of intra- and extracellular viral RNA load at different time points (24, 48, and 72 h p.i) using RT-qPCR. The intracellular levels of viral RNA are already high in the first 24 h, increasing over 48 h and declining within 72 h of infection. Nonetheless, viral RNA in the cell supernatant increased with time after infection and remained practically constant.

### 3.4. Antiviral Activity of Uncaria tomentosa against CHIKV Determined by RT-qPCR and Flow Cytometry

To assess the UT antiviral effect on CHIKV infection, Vero cells were CHIKV infected at low MOI (0.01) and then treated with different non-cytotoxic extract concentrations observed through MTT testing and 0.4% of vehicle control (DMSO). After 24 h p.i, RNA levels detected in cell supernatant showed a significant decrease, and UT induced a reduction of 93% at 100 µg/mL as compared to CHIKV-positive control and UT-treated cells at 6 µg/mL. Indeed, a decrease in viral RNA was observed in UT-treated Vero cells as we observed a significant negative linear trend at 24 h (*p* < 0.0001) and 48 h p.i (*p* = 0.0025), respectively. At 72 h p.i, UT treatment caused no significant reduction in CHIKV RNA copies (Figure 3A).

Similarly, 100 µg/mL of UT was effective (approximately90%) in reducing intracellular viral RNA at 24 h as compared to CHIKV-infected cells treated with vehicle control. Additionally, a significant negative linear trend was observed during 24 h p.i (*p* = 0.0054) (Figure 3B). The CHIKV has a characteristic of producing cytopathic effect (CPE) in susceptible vertebrate cells, and we observed that a dose of 100 µg/mL of UT was able to inhibit the CPE at 48 h p.i (Figure 3C).

Finally, the EC50 calculated for the extract by RT-qPCR after 24 h of treatment was 19.85 µg/mL (CI95% interval of 9.6 to 32.5 µg/mL).

After demonstrating the antiviral effect of UT extract against CHIKV infection, especially with 50 µg/mL and 100 µg/mL at 24 h p.i, we evaluated the intracellular expression of CHIKV antigen in infected and treated Vero cells at 24 h p.i to confirm this finding. For this, we performed flow cytometric analysis for quantification of treated and untreated CHIKV-infected cells (Figure 4A). Strategy for the CHIKV-infected cells in flow cytometric experiments is shown in Appendix A. As demonstrated in Figure 4B,C, the percentage of CHIKV-positive cells was significantly reduced when infected cells were treated with 50 µg/mL and 100 µg/mL of UT as compared to CHIKV-positive control and CHIKV-infected cells treated with DMSO (vehicle control). Also, Vero cells treated at 100 µg/mL showed a statistically significant decrease in the percentage of positive cells compared to those treated at 50 µg/mL. The data showed a statistically significant difference in mock-treated cells compared to CHIKV-positive and vehicle control as well. 

### 3.5. Antiviral Effect of Uncaria tomentosa Extract Determined by Plaque Assay

Given the results obtained in the antiviral activity investigation determined by RT-qPCR and flow cytometry, concentrations of 50 µg/mL and 100 µg/mL were selected to evaluate the reduction in CHIKV infectivity by plaque reduction assay after 24 h and 48 h of treatment. In the first 24 h of incubation, treatment at 50 and 100 µg/mL reduced viral load to 7.16 × 10^5^ PFU/mL and 1.93 × 10^5^, respectively, which represent 89% and 97% inhibition (Figure 5A), as positive control showed 6.69 × 10^6^ PFU/mL. Indeed, a significant negative linear trend at 24 h p.i (*p* = 0.0317) was observed (Figure 5B). Despite no statistical significance, after 48 h of incubation, a concentration of 100 µg/mL showed an inhibitory effect, reducing the viral load from 1.49 × 10^7^ PFU/mL of positive control to 4.69 × 10^6^ PFU/mL in treated cells supernatant, representing 68% reduction (Figure 5C). The plaque reduction assay findings confirmed the viral RNA quantification in cell supernatants, indicating that 100 µg/mL induces a better inhibitory effect than 50 µg/mL, and it is more effective in the 24 h p.i.

### 3.6. Virucidal Effect of Uncaria tomentosa Extract Determined by RT-qPCR

In addition to the antiviral effect, we also explored the possibility of the UT extract inactivating CHIKV particles (virucidal effect). As described in the material and methods, CHIKV was incubated for 1 h at 37 °C and 5% CO_2_ at 50 and 100 µg/mL of UT extract. Subsequently, Vero cells were infected with the treated virus, and the cell supernatant was tested for RNA virus quantification. As represented in Figure 6A, the CPE on Vero cells after 48 h p.i is apparent, and no CPE was detected in CHIKV-infected Vero cells treated at 100 µg/mL of UT. Although no statistically significant changes were observed after 24 h p.i, treatment at 50 µg/mL and 100 µg/mL reduced CHIKV RNA copies/mL by 95% and 98%, respectively. In addition, a statistically significant decrease in RNA copies was observed after 48 h p.i in the treated virus at 100 µg/mL as compared to the CHIKV-positive control. Furthermore, a statistically significant negative linear trend was observed, indicating a decrease in RNA copies after 48 h p.i (*p* = 0.0071). No statistically significant changes were observed after 72 h p.i (Figure 6B).

### 3.7. Virucidal Effect of Uncaria tomentosa Extract Determined by Plaque Assay

Since we detected the UT virucidal effects against CHIKV, the 100 and 50 µg/mL concentrations were tested using plaque reduction assay. Supernatants were collected at 24 h and 48 h p.i, and the viral load was titrated. Confirming the RT-qPCR findings, treatment with 100 µg/mL caused a significant reduction in viral infectivity at 24 h p.i with a 98% decrease in viral load (Figure 6C,D). The difference between CHIKV-positive control and UT treatment was not statistically significant after 48 p.i. However, there was a 75% reduction in viral load (Figure 6E). These results revealed that hydroalcoholic extract of UT at a concentration of 100 µg/mL also showed virucidal activity against CHIKV-infected Vero cells in vitro.

## 4. Discussion

The simultaneous circulation of emerging and re-emerging viruses such as the new coronavirus, Severe Acute Respiratory Syndrome (SARS-CoV-2), and arboviruses is a major concern in Brazil [45]. Notably, CHIKV quickly spread to new geographic regions, causing outbreaks and explosive epidemics in Brazil [46]. The evolution and spread of CHIKV were marked by mutations in the viral envelope glycoproteins E1 and E2, making these proteins key determinants of infectivity and pathogenesis essential for viral adaptation [17,47]. According to the Ministry of Health, in the first semester of 2023, 143.739 cases of the disease were reported in Brazil, with an incidence rate of 67.4 cases per 100 thousand inhabitants in the country [48]. Studies aimed at solutions such as the development of vaccines and antiviral treatments are urgent, but unfortunately, research in these areas is still scarce. The Valneva VLA1553 chikungunya vaccine induces an immune response in 98.8% of those vaccinated, according to the American data from the phase 3 clinical trial published recently [49]. However, there is still no expected release date for use in Brazil. In this sense, we highlight the urgency of studies that aim to provide emergency responses to CHIKV infection and that will contribute to moving towards alternative therapeutic methods, such as the use of natural products.

Chikungunya fever is an acute febrile illness that causes fever, myalgia, headache, and other unspecific symptoms, but it is characterized mostly by its intense joint pain [15,16]. Post-chikungunya arthritis is the most common clinical manifestation after infection, and it is estimated that 40 to 60% of cases have significantly impaired long-term quality of life after infection. A systematic review of post-chikungunya chronic inflammatory rheumatism estimates a pooled prevalence of 40% or 25%, depending on the selected parameters [50]. Beyond arthritogenic manifestations, whether due to improvements in diagnosis and case reporting or to mutations in the viral genome, neonatal and children infection has been reported more frequently since its re-emergence in 2005. Regardless of the reason, CHIKV infection in this age group causes cardiac defects such as myocardial hypertrophy, ventricular dysfunction, and pericarditis [51]. In the central nervous system, meningoencephalitis, microcephaly, and developmental delay were reported in children, too [51,52].

Despite the serious and debilitating symptoms, up to now, no antiviral drug against CHIKV has been approved, and treatment is based on symptom relief. WHO guidelines recommend acetaminophen or paracetamol to relieve fever and non-steroidal anti-inflammatory drugs (NSAIDs) to reduce joint pain during the acute phase. Regarding the chronic phase, there is a lack of consensus depending on the source, as reviewed by Webb et al. [53]. Disease-modifying anti-rheumatic drugs (DMARDs) such as NSAIDs, corticosteroids, methotrexate, and hydroxychloroquine are prescribed, depending on pain intensity and response to treatment. However, hydroxychloroquine efficacy in chikungunya cases is questioned by several studies [54,55,56]. Methotrexate, an antifolate, is a promising drug for chronic chikungunya arthritis, but more studies are necessary since the treatment is not always effective and the methodology and posology vary [57,58,59]. In this scenario, it is essential to discover therapeutic alternatives for disease control and better quality of life for acute and chronic patients.

Medicinal plants and natural compounds have been investigated for the treatment of arboviruses [60,61,62]. In addition to the results already reported for the *Uncaria* genus [34,35,36], recently, we showed the potential of the species *Miconia albicans* as a chikungunya fever treatment. This species is consumed all over the Brazilian territory as a remedy to treat rheumatoid arthritis and has already been increasingly used to alleviate the deleterious symptoms caused by CHIKV [63].

Antiviral properties of the *Uncaria* genus have been previously communicated [34,35,36,64]. The hydroalcoholic extract of UT bark was able to reduce herpes simplex virus-1 (HSV-1) infection in Vero cells. Possibly, its effect is related to the simultaneous action of compounds present in the extract since the purified fraction of quinovic acid glycosides and oxindole alkaloids did not show an antiherpetic effect [64]. Regarding the in vivo effect, the extract containing 5% mitraphylline has been suggested as a treatment for cold sores, but due to the anti-inflammatory effect, characteristic of the species [65]. UT also showed an intense inhibitory effect in SARS-CoV-2 plaque formation and cytopathic effect in Vero cells, reaching 92.7% and 98.6% reduction, respectively, when cells were treated with 25 µg/mL for 48 h [66].

In the search for natural products that could have therapeutic properties against dengue, we obtained promising results with UT and UG extracts on viral clearance and immunomodulation with the reduction of cytokines related to the poor prognosis of dengue [34,35,36]. Our previous study demonstrated that in human monocytes infected with DENV-2, intracellular viral antigen detection was reduced after 10 µg/mL of hydroalcoholic extract treatment at 48 h p.i. and its alkaloidal fraction showed an inhibitory effect against DENV-2 infection [34]. Alkaloidal fraction also inhibited DENV-2 infection in a model of human lineage of dermal microvascular endothelial cells (HMEC-1), reducing NS1 detection in cell supernatant after 24 h treatment with 1 µg/mL and 48 h treatment with 1 and 10 µg/mL [35]. We confirmed the antiviral effect of UT hydroalcoholic extract previously found against DENV infection. UT hydroalcoholic extract effectively reduced intracellular CHIKV RNA and the release of extracellular CHIKV particles at 24 and 48 h p.i. An intense decrease in CHIKV RNA was detected in cell supernatant after 24 and 48 h p.i treatment with 50 and 100 µg/mL as well. Indeed, at an early point, the intracellular CHIKV antigen was reduced at 50 and 100 µg/mL, confirming the RT-qPCR results. Importantly, UT treatment was able to protect Vero cells from CHIKV-induced CPE. At 72 h p.i, no antiviral effect was detected. This observation might be related to the reduction in the number of viable cells over the three days evaluated, considering the intense cytopathic effect of CHIKV in mammal cells [42,67].

RT-qPCR is a widely used technique in antiviral research, given its high sensitivity and specificity to quantitate virus RNA. Our study uses a previously described protocol with sensitivity to detect 0.3 PFU [40] as a tool to detect CHIKV RNA. Simultaneously, we tested the virus infectivity using plaque assay, the gold standard for quantification of lytic virus [68] that confirmed CHIKV infectivity reduction after 24 h and 48 h treatment with 100 µg/mL.

Once the antiviral activity was observed, we investigated the capacity of the extract to inactivate the viral particle, in other words, a direct effect on the viral particle, the virucidal effect. After pretreatment of CHIKV with UT at 100 µg/mL, we observed greater inhibition capacity, reducing viral RNA detection by around 90% after 24 and 48 h p.i. In addition, the plaque assay demonstrated virucidal activity with UT CHIKV treated at 100 µg/mL after 24 h p.i. To the best of our knowledge, this is the first report of UT antiviral and virucidal effects against CHIKV infection.

The susceptibility of Vero cells to the chikungunya virus is widely known, and the fast kinetics and intense growth observed here are confirmed by other authors. Vero, HuH-7, and A549 cells have been used as models to investigate the action of broad-spectrum antivirals on CHIKV. The replication pattern, which authors evaluated using the plaque assay, was similar to that observed in our study, increasing over the 3 days of infection [69]. A previous study reported crescent viral loads measured by TCID50 in Vero E6 cells, too, with a maximum viral load of 8 logs on day three [70]. This phenomenon can also be explained by the ability of the virus itself to produce high viral loads, either in patients [71] or in vitro models, such as in epithelial cell lines HeLa, 293T, BEAS-2B, and in primary fibroblasts MCR5 [44].

Maneuvering different approaches, we demonstrated that the UT-hydroalcoholic extract has antiviral and virucidal activity against CHIKV in an in vitro infection model using Vero cells. The exact mechanism by which UT inhibits viral infection in cells is unknown. In SARS-CoV2 studies, it was demonstrated, in silico, that three different compounds present in the hydroalcoholic extract are capable of interacting with the CLpro protein [72]. In addition to the interaction with the viral protein, the authors discovered a good therapeutic behavior of these compounds by measuring their absorption, distribution, metabolism, and excretion (ADME-score), indicating the therapeutic potential of the plant [72]. Maceration extract of pure UT and that combined with UG had an antiherpetic effect in the Vero cell model, possibly inhibiting viral attachment, in a simultaneous action of compounds present in the extract since the purified fractions did not show an antiherpetic effect [64]. Our results indicate that UT may act directly in viral particles or at the beginning of CHIKV infection, too, but more studies are necessary to elucidate the mechanism.

It is also important to acknowledge that our study has certain limitations. Although it is a widely used model in studies with arboviruses, Vero cells do not produce interferon [73,74]. More studies using in vitro models of human cells are needed for a better representation of human infection and a better understanding of the antiviral activity of the extract. In addition, new models of infection will also allow evaluation of the immunomodulatory activity during CHIKV infection. Indeed, we demonstrated the inhibitory properties of UT in human monocytes infected with CHIKV (Unpublished data). Our results indicate that the extract is effective in the first 48 h of infection. Other assays at shorter intervals of infection, such as 6 h or 8 h might show more intense inhibitory effects, as well as treatments in sequence, to reset the biologically active substances present in the extract. It is also important to highlight that the effects obtained in vitro are not always reflected in vivo, which reinforces the need for more studies to confirm the potential of UT as a treatment for chikungunya fever.

Chikungunya remains a public health problem, causing epidemics around the world. Despite causing intense and debilitating pain, there is still no antiviral drug or unified treatment protocol for chronic cases, which makes it essential to search for new alternatives to fight infection and the chronicity of the disease. Here, we show that the medicinal plant *Uncaria tomentosa* exhibits potential antiviral and virucidal activity against CHIKV, reducing the number of viral RNA copies, percentages of CHIKV antigen, and infective particles in CHIKV-infected Vero cells in vitro. Collectively, the data presented in this study provide further evidence as a potential therapeutic option against alfaviruses.

## Figures and Tables

**Figure 1 viruses-16-00369-f001:**
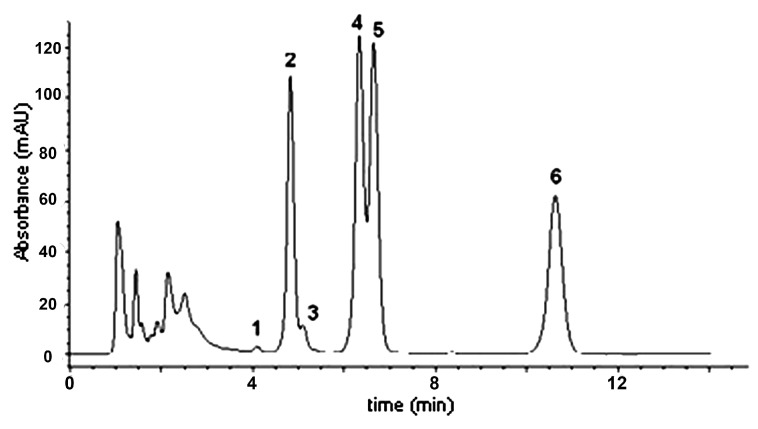
Pentacyclic oxindole alkaloid HPLC-UV profile of the studied *Uncaria tomentosa* extract assayed: 1: Speciophylline; 2: Mitraphylline; 3: Uncarine F; 4: Pteropodine; 5: Isomitraphylline; 6: Isopteropodine (Figure from [41] with CC-BY attribution).

**Figure 2 viruses-16-00369-f002:**
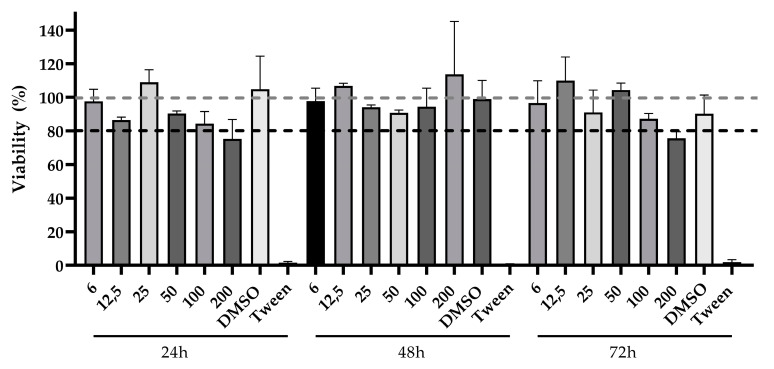
Cytotoxicity of *Uncaria tomentosa* extract in Vero cell line cultures. Vero cell viability was assessed by MTT assay after 24 h, 48 h, and 72 h treatment with UT hydroalcoholic extract at different concentrations. The highest DMSO concentration (0.4%) was included as vehicle cytotoxicity control, and Tween at a concentration of 3% was included as cytotoxicity control. Untreated cells were used as negative control and considered as a reference of 100% viability. Three independent experiments were tested with 3–6 replicates each.

**Figure 3 viruses-16-00369-f003:**
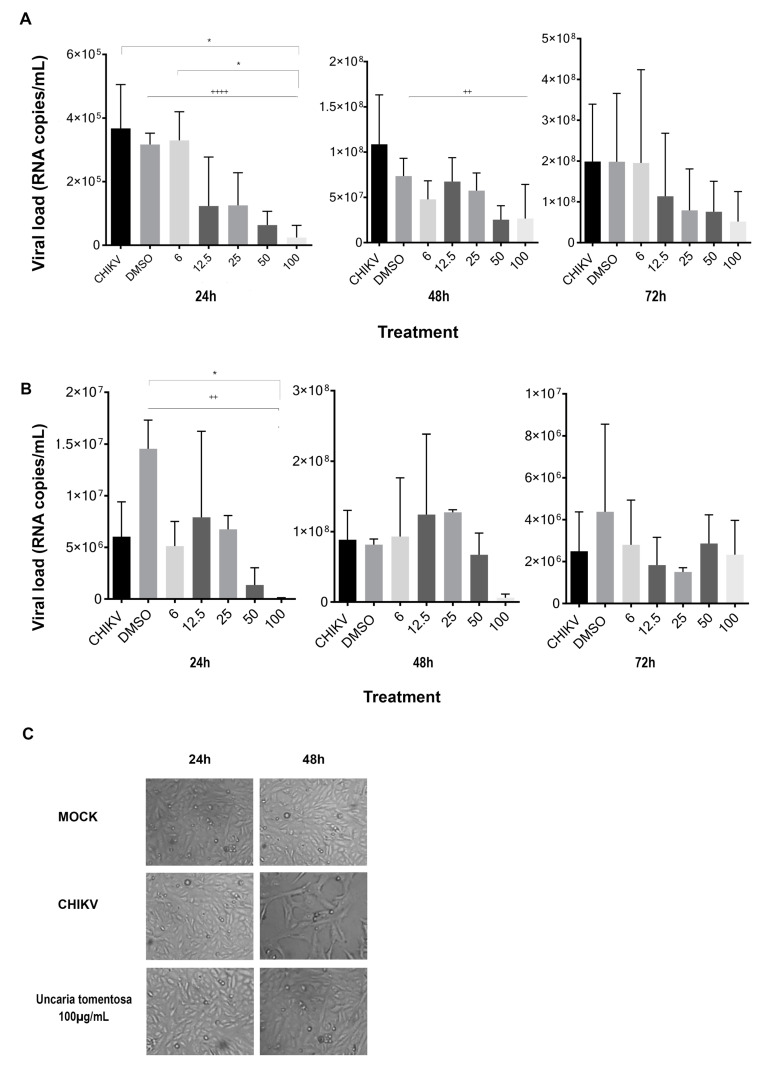
Antiviral activity of *Uncaria tomentosa* against CHIKV infection. (**A**) Extracellular and (**B**) Intracellular CHIKV RNA quantification by RT-qPCR of CHIKV-infected Vero cells at an MOI of 0.01 and treated with different concentrations (6–100 µg/mL) of hydroalcoholic extract of UT for 24, 48 h, and 72 h. The data represent the means and standard deviation (±SD) of three independent experiments in duplicates. * *p* < 0.05, one-way ANOVA followed by Turkey test. Linear regression analysis after conversion to logarithm of the doses, significant ++ *p* < 0.01, ++++ *p <* 0.0001 indicated. (**C**) Cell morphology of Vero cells at 24 and 48 h p.i. Representative images of negative control (Mock), positive control (CHIKV infected), and CHIKV-infected cells after treatment with 100 µg/mL.

**Figure 4 viruses-16-00369-f004:**
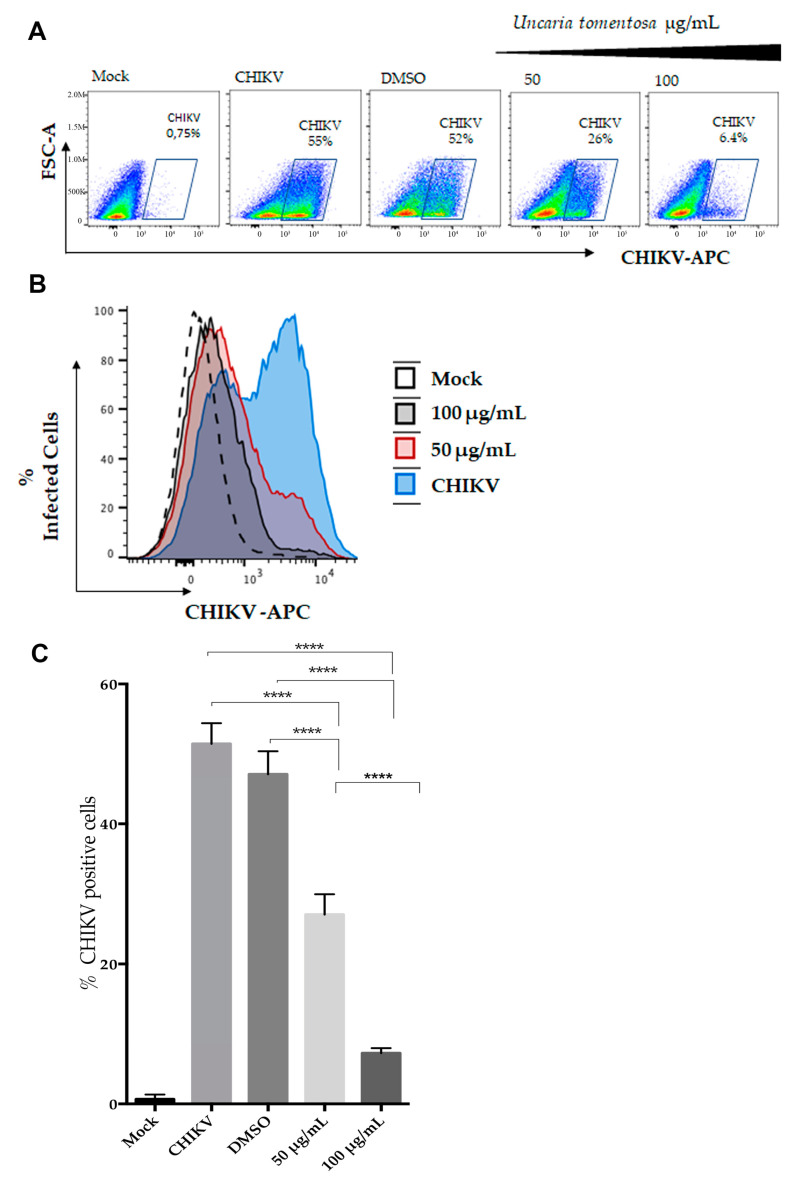
Flow cytometric analysis for quantification of treated and untreated CHIKV-infected cells. (**A**) Representative dot plot analysis (FSC × CHIKV-Alexa-647). The CHIKV-positive cells are represented in the gates. From left to right, mock (negative control), CHIKV (positive control), DMSO (vehicle control), 50 µg/mL, and 100 µg/mL of UT are represented. (**B**) Histogram analyses showing the percentage of positive cells for each condition: negative control (Mock, dashed line), CHIKV (positive control, blue), and UT at 50 µg/mL (red) and 100 µg/mL (gray), respectively. (**C**) Percentage of positive cells determined by flow cytometry. Two independent experiments in duplicates. Data are expressed as mean (±SD). **** *p* < 0.0001, one-way ANOVA followed by Turkey test.

**Figure 5 viruses-16-00369-f005:**
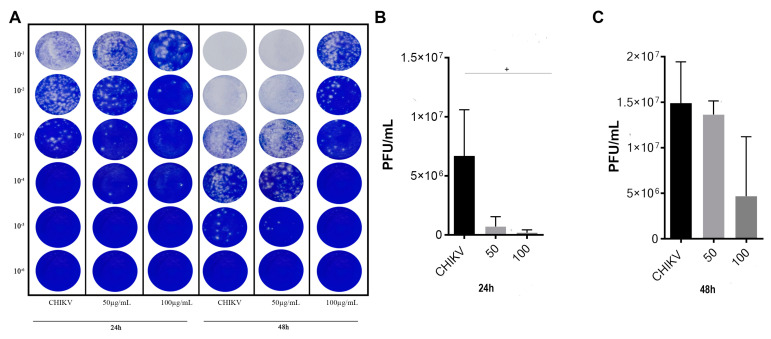
Antiviral effect of *Uncaria tomentosa* extract determined by plaque assay. (**A**) Representative image showing plaques of the antiviral activity of the UT extract against CHIKV infection. (**B**) Quantification of Viral titer (PFU/mL) in the supernatants of infected Vero cells after treatment with UT extract (50 and 100 µg/mL) by plaque assay (24 h p.i). (**C**) Quantification of viral titer (PFU/mL) in the supernatants of infected Vero cells after treatment with UT extract (50 and 100 µg/mL) by plaque assay (48 h p.i). Linear regression analysis after conversion to logarithm of the doses, significant + *p* < 0.05 indicated.

**Figure 6 viruses-16-00369-f006:**
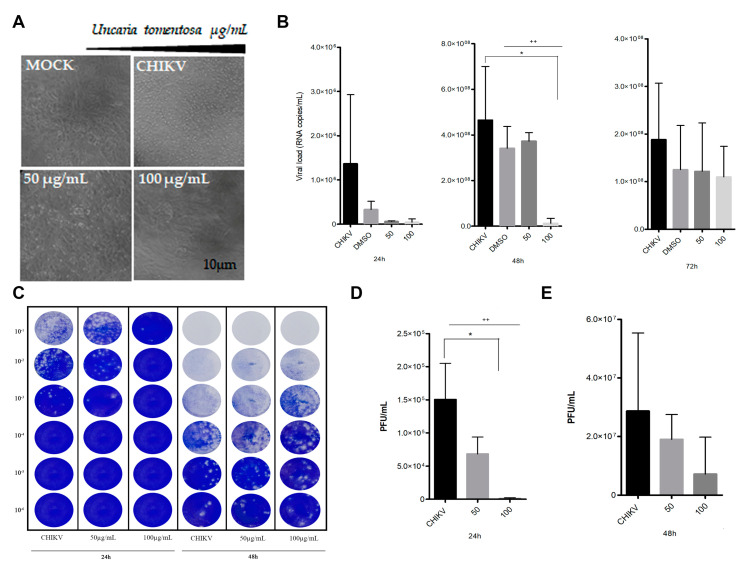
Virucidal effect of *Uncaria tomentosa* against CHIKV infection. (**A**) Virucidal effect of UT showing cell morphology at 48 h post-infection. Representative images of negative control (Mock), positive control (CHIKV infected), and CHIKV-infected cells after treatment with 50 and 100 µg/mL, respectively. (**B**) Quantification of viral titer (RNA copies/mL) in the supernatants of infected Vero cells treated at 50 and 100 µg/mL by RT-qPCR after 24 h, 48 h, and 72 h p.i. (**C**) The virucidal effect of *Uncaria tomentosa* determined by plaque assay. Representative image showing plaques of UT virucidal activity against CHIKV-infected cells. Quantification of viral titer (PFU/mL) in the supernatants of infected cells treated at 50 and 100 µg/mL, (**D**) after 24 h and (**E**) 48 h p.i by plaque assay. * *p* < 0.05, ANOVA followed by Tukey. Linear regression analysis after conversion to logarithm of the doses, significant ++ *p* < 0.01 indicated.

## Data Availability

All the data available is included in the manuscript.

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
