# Peer review of "Antiviral and Virucidal Activities of Uncaria tomentosa (Cat’s Claw) against the Chikungunya Virus"

_viruses, 2024, doi:10.3390/v16030369_

Round 1

Reviewer 1 Report

Comments and Suggestions for Authors

In this manuscript, the authors evaluated the antiviral effect of Uncaria tomentosa against Chikungunya virus in vero cells. They nicely showed that UT is able to inhibit CHIKV replication by RT-PCR, flow cytometry and plaque assay at early time points. However, the statistical analysis needs more specification and validation in another cell type is recommended.

Key concerns to be addressed are listed below:

1.     The statistical analysis throughout this paper is a bit confusing in general. Can author explain what is “Post-test” and the groups they involve for this analysis (to be specific, does authors only conduct post-test in UT-treated groups)? If possible, please add reference for this post-test which will be helpful for the audience.

In addition, the presentation of statistical result need improvement as well. Please specify the p-values

2.     In Figure 2, can authors mark the line of 100% viability as well for clearer visualization and comparison?

3.     In Figure 3, (1) the fonts are extremely small and very hard to read, please make modifications so it is easier to see.

(2) the figure legend for 3A and 3B are highly repetitive and have a lot of redundant information. Please simplify the description.

(3) the p-values provided in 3A(0.031) and 3B(0,031) do not match the p-value marked in the figure. Please explain and better specify the scenario for each p-value provided in the figure since there are multiple tests performed.

4.     Line 298, should be 19.85 instead of 19,85.

5.     In Figure 4, please fix the title, the current one doesn’t read properly.

6.     Line 385, should be 143,739 instead of 143.739.

7.     A general take-home information from this paper is that the UT doesn’t have significant antiviral effect after 72 hours post infection. Is it possible that UT lost its activity over time? Have the author tried adding UT back every 24 hour to see if it can retain it anti-CHIKV activity?

8.     One major concern is that this paper only tested the antiviral effect of UT in one cell-type which is a monkey cell line are doesn’t have any interferon response, if the authors can validate the result in another clinically-relevant cell line, it will add scientific soundness to the result.

9.   Please add a brief discussion about the potential mechanism of UT.

Reviewer 2 Report

Comments and Suggestions for Authors

The article describes a potential antiviral and virucidal activity of Uncaria tomentosa (UT) against Chikungunya virus. Although the overall idea and importance of this research are relevant, some experiments need to be improved before publication. 

First point - text need improvement. Lines 42 and 43, list the two groups; line 80 - Is uña de gato most used than unha de gato? All experiments were conducted in Brazil, but the name of this plant is in Spanish; 

Methodology - All experiments were done in Vero cells, and the authors themselves affirm that this is not the ideal cell line. I would advise using a human cell line well suited to CHIKV infections. Although the authors mentioned that some experiments were done using monocytes (data not shown), the virus does not replicate well in these cells. Another methodology point that I was confused about was the number of cells seeded per well. In a 96-well plate 105 cells were seeded per well and incubated for 24 hours. This is a huge number of cells per well on a 96-well plate, especially for Vero cells. Please, check the numbers. For the 12-well plate, the number of seeded cells were 2x105 per well, which can be ideal but it shows a big discrepancy with 96 well plate. Another point that I would change is the analytical sensitivity of the qPCR test, which was given in 0.3 PFU, but I would use the number of RNA copies. PFU reflects infectious particles, but PCR can amplify free RNA (cDNA) or even genetic material for non-infectious particles.

I am not sure if the pdf version had some mistakes with some legends, but graph legends have typos or are in a weird presentation.

Figure 2 - Y axis - I believe that this axis could be improved and present the 100% to make data visualization easier. Also, DMSO, which sometimes is used as a positive control, had more viability than non-treated cell control. A little bit strange, but maybe the low concentration did not reflect any effect on these cells (0.4%). The legend of this figure says that untreated cells were positive controls, I would call them negative controls.

Figure 3. Although the authors pointed out a possible antiviral activity of UT against CHIKV, this effect was observed only at 24hpi. I am not aware if this compound is being degraded or if this effect could be improved with sequential doses of UT.

Figure 4B - Difficult to see, I would change colors to make data visualization easier for the readers. 

The discussion is very long, most paragraphs could be placed in the introduction, and little is discussed about the experiments that were done. Also, the authors conclude that this compound has antiviral and virucidal activity, but it has potential, only in vitro studies with a non-human cell line were performed. Several compounds can show in vitro activity against viruses, but that is not reflected in animal models or clinical studies.

Round 2

Reviewer 1 Report

Comments and Suggestions for Authors

I appreciate the author's efforts to revise this manuscript, and I believe it has been significantly improved compared to last version. With some minor revisions, I think this paper is suitable for publication at Viruses

1.  In Figure 3A, the font size in first panel is smaller than the other 2 panels, please fix. In Figure 3B, fist panel, should be 12.5 instead of 12,5.

2. Line 359,  48h p.i instead of 48 p.i.

3. In Figure 6B, please fix the number in the y axis of the 2nd and 3rd panel so they correlate with the rest of the figure. For example, 1.0 x 108 instead of 1.0 x 1008, etc.

4. Line 325, p=0.0317 not p=0,0317.

5. Line 381, is it 143,739 cases?

Reviewer 2 Report

Comments and Suggestions for Authors

The authors made substantial improvements in this manuscript and, in my opinion, it is good for publication.